# Serum D-Lactate Concentrations in Dogs with Inflammatory Bowel Disease

**DOI:** 10.3390/ani14111704

**Published:** 2024-06-06

**Authors:** Giulia Maggi, Elisabetta Chiaradia, Alice Vullo, Matteo Seccaroni, Laura Valli, Sara Busechian, Domenico Caivano, Francesco Porciello, Sabrina Caloiero, Maria Chiara Marchesi

**Affiliations:** 1Department of Veterinary Medicine, University of Perugia, Via San Costanzo 4, 06126 Perugia, Italy; giuliamaggi217@gmail.com (G.M.); elisabetta.chiaradia@unipg.it (E.C.); alice.vullo@studenti.unipg.it (A.V.); matteo.seccaroni1@studenti.unipg.it (M.S.); sarabusechian@gmail.com (S.B.); francesco.porciello@unipg.it (F.P.); 2Independent Researcher, 23891 Rome, Italy; lauravalli.vet@gmail.com; 3Kennel Training Course Castiglione del Lago of Financial Guard, Via Lungolago 46, 06061 Castiglione del Lago, Italy; caloiero.sabrina@gdf.it

**Keywords:** canine, D-lactate, dysbiosis, gastroenterology, inflammatory bowel disease

## Abstract

**Simple Summary:**

In humans and animals, the D-enantiomer of lactic acid (D-lactate) is normally produced from bacterial fermentation in the gastrointestinal tract. During gastroenteric diseases, D-lactate can be produced in large quantities and absorbed by the intestinal mucosa. The purpose of the present study was to measure the serum D-lactate concentrations in dogs with chronic inflammatory bowel disease (IBD). For this reason, the serum D-lactate concentrations were measured in 10 healthy dogs and 18 dogs with IBD using a commercially available colorimetric assay kit. Our results showed no significant difference (*p* > 0.05) in the serum concentrations of D-lactate between dogs with various degrees of IBD and healthy dogs. Further studies are needed to understand potential factors able to influence the serum D-lactate concentrations in dogs affected by IBD.

**Abstract:**

The D-enantiomer of lactic acid (D-lactate) is normally produced from bacterial fermentation in the gastrointestinal tract in mammals. In humans, increased D-lactate concentrations are related to gastrointestinal disease, including short bowel syndrome and malabsorptive syndrome. Similarly, increased D-lactate concentrations have been described in calves affected by diarrhea, in cats with gastrointestinal diseases, and in dogs with parvoviral enteritis. The purpose of the present study was to measure the serum D-lactate concentrations in dogs with inflammatory bowel disease (IBD). We retrospectively reviewed data from the database of the VTH of Perugia University, and dogs affected by IBD with serum samples stored at −80 °C were considered eligible for inclusion. A total of 18 dogs with IBD and 10 healthy dogs were included in the study. The dogs with IBD were divided into three subcategories based on the severity of the disease. Serum D-lactate concentrations (μM) were determined using a commercially available colorimetric assay kit (D-Lactate Colorimetric Assay Kit; Catalog #K667-100, BioVision Inc., Milpitas, CA, USA). Our results showed no significant difference (*p* > 0.05) in the serum concentrations of D-lactate between dogs with various degrees of IBD and healthy dogs. However, the wide variability of the D-lactate concentrations in dogs with IBD and evidence of increased serum D-lactate concentrations in dogs with confirmed dysbiosis encourage further studies on this topic to understand potential factors influencing the serum D-lactate concentrations in dogs affected by IBD.

## 1. Introduction

The D-enantiomer of lactic acid (D-lactate) is minimally produced endogenously via the methylglyoxal pathway but is typically generated from exogenous sources, such as bacterial fermentation of carbohydrates in the gastrointestinal tract in mammals [1,2,3]. For these reasons, D-lactate is not commonly detectable in the serum of these species, and its elevation is considered specific to bacterial origin [1,3]. In humans, increased serum D-lactate concentrations are associated with gastrointestinal diseases, including short bowel syndrome and malabsorptive syndrome [4]. Similarly, increased D-lactate concentrations have been well documented in diarrheic calves and can also occur in cats with gastrointestinal disease, diabetic ketoacidosis, and propylene glycol ingestion [5,6,7,8,9]. In dogs, there is a proton-linked monocarboxylate transporter (MCT1) for the systemic absorption of D-lactate in the colon. Higher concentrations of fecal D-lactate have been observed in dogs with chronic enteropathies (CEs) and exocrine pancreatic insufficiency (EPI) [10]. Only a few studies have measured the concentrations of serum D-lactate in dogs with gastrointestinal diseases, showing D-lactate acidosis in dogs affected by parvoviral enteritis [3,11].

Idiopathic inflammatory bowel disease (IBD) in dogs is defined as a syndrome characterized by chronic gastrointestinal (GI) signs lasting more than 3 weeks, with the presence of mucosal inflammation but without overt evidence of an identifiable etiologic agent or causative factor [12]. The pathogenesis of canine IBD remains undetermined. It is considered a multifactorial immune-mediated disease, resulting from complex interaction between environmental factors, the microbiome in genetically susceptible individuals, and the mucosal immune system [13]. CEs include disorders characterized by persistent or recurrent GI signs and clinically can be classified into food-responsive (FRE), antibiotic-responsive (ARE), immunosuppressant-responsive enteropathy (IRE), and non-responsive enteropathy (NRE) [14,15,16]. Recent studies have identified alterations in various bacterial groups within the intestinal microbiome in dogs with CEs compared to healthy dogs [17,18,19]. The total bacteria, *Faecalibacterium*, *Turicibacter*, *Blautia*, *Fusobacterium* and *Clostridium hiranonis* levels were significantly lower in dogs with CEs compared to healthy dogs. Conversely, the *Escherichia coli* and *Streptococcus* levels were significantly higher in affected dogs [19]. Similarly, microbiome alterations were observed in dogs with EPI, along with a significant increase in fecal lactate (both D-lactate and L-lactate) [10]. In human patients affected by short bowel syndrome, excess carbohydrates undergo bacterial fermentation, increasing the D-lactate concentrations within the colon. This results in an acidic environment, where acid-resistant bacteria such as *Lactobacillus* and *Streptococcus* proliferate [4,20]. Additionally, alterations in the fecal microbiome were reported in diarrheic calves with D-lactate and L-lactate acidosis as compared to healthy calves [21]. A clear relationship has emerged between increased concentrations of D-lactate in the gut and the bacterial species comprising the gut microbiome in both humans and animals [4,10,20,21]. To date, microbiome changes in dogs with IBD can be evaluated using a mathematical algorithm known as the dysbiosis index (DI), which is based on the results of quantitative PCR (qPCR) assays performed on fecal samples [19]. However, due to the costs and limited availability of qPCR for the DI in veterinary clinical practice, new markers of dysbiosis in dogs with IBD are needed.

The purpose of the present study was to measure the serum D-lactate concentrations in dogs with IBD. Our hypothesis was that the dogs with IBD would exhibit increased serum D-lactate concentrations, potentially as a consequence of dysbiosis, compared to healthy dogs.

## 2. Materials and Methods

### 2.1. Animals

We conducted this retrospective observational study at the Veterinary Teaching Hospital (VTH) of the University of Perugia. Using the VTH database, a group of dogs with IBD was preliminarily selected. The inclusion criteria for dogs selected for a definitive diagnosis of IBD were based on (1) a chronic history (at least 3 weeks) of GI signs, such as anorexia, reduced appetite, vomiting, diarrhea, and weight loss; (2) exclusion of food-responsive enteropathy (FRE); (3) negative response to therapy with probiotics; (4) negative results of diagnostic tests ruling out other disorders associated with chronic GI signs, including mechanical GI obstruction, GI neoplasia, EPI, hepatic disease, endocrinopathies, pancreatitis, and pancreatic tumors; (5) confirmation of intestinal inflammation by histological findings. The diagnostic procedures provided for each dog included a complete blood count and serum biochemical profile, serum trypsin-like-immunoreactivity (TLI), serum folate, serum cobalamin, basal cortisol, and abdominal ultrasound. A diet trial with hydrolyzed food for at least 4 weeks was performed to exclude food-responsive enteropathy (FRE). Additionally, a multi-strain probiotic (VSL3#, Ferring Spa, Milan, Italy) was administered at one packet (4.4 g) q24 h for 4 weeks to exclude CEs responsive to modulation of the microbiome. Dogs with EPI or hypoadrenocorticism and those showing a positive response to the diet trial or probiotic therapy were excluded from the study. Because the purpose of our study was to measure the serum D-lactate concentrations in dogs with IBD, only dogs with serum samples stored at −80 °C were considered eligible for inclusion.

The dogs with IBD were divided into three subcategories in relation to severity of the disease. For these reasons, a scoring “Index for IBD activity” (Table 1) was created based on the Canine Chronic Enteropathy Activity Index (CCECAI) [22], the World Small Animal Veterinary Association (WSAVA) histopathological grading [13], and the concentrations of serum albumin (Alb). The “Index for IBD activity” used in this study had a total score of 10, with the severity of IBD classified as mild for a score of 3 or less, moderate for a score between 4 and 6, and severe for a score of 7 or higher. Based on the WSAVA histopathological grading and serum Alb concentration, scores ranging from 0 to 3 were attributed to each class. Concerning the WSAVA histopathological grading, we determined the severity of the inflammation (mild, moderate and severe) for each section of the gastrointestinal tract (stomach, duodenum, colon, and ileum). To determine the final score included in the “Index for IBD activity”, we considered the highest severity among the different sections examined. Additionally, based on the CCECAI, scores ranging from 0 to 4 were attributed. The “Index for IBD activity” variables were calculated as follows: (1) the CCECAI (0 = Normal 0–3, 1 = Mild 4–5, 2 = Moderate 6–7, 3 = Severe 9–11, 4 = Very Severe > 12); (2) the WSAVA histopathological grading (0 = Normal, 1 = Mild, 2 = Moderate, 3 = Marked); (3) Alb concentrations (0 = > 2 mg/dL, 1 = 1.5–1.9 mg/dL, 2 = 1.2–1.49 mg/dL, 3 = < 1.2 mg/dL). To determine the total score, we summed each variable.

To obtain a group of clinically healthy dogs (control group), animals from the VTH blood donor program were reviewed. Dogs with serum samples stored at −80 °C were included. Serum samples were obtained concurrently with screening checks, which were conducted independently of our study and were necessary to assess the health status of the animals. The blood donor dogs followed the criteria outlined in the national Ministry of Health Guidelines. Each animal was considered clinically healthy based on its thorough clinical history, physical examination, complete CBC, and biochemical and urinalysis profile carried out as part of routine clinical screening. Additionally, the donor dogs were confirmed to be free from blood-borne diseases through negative serological tests for *Anaplasma phagocytophilum*, *Babesia* spp., *Dirofilaria immitis*, *Ehrlichia canis*, and *Leishmania infantum*.

### 2.2. D-Lactate Measurements

The serum D-lactate concentrations (μM) were determined using a commercially available colorimetric assay kit (D-Lactate Colorimetric Assay Kit; Catalog #K667-100, BioVision Inc., Milpitas, CA, USA). This assay relies on the oxidation of D-lactate by D-lactate dehydrogenase to generate a proportional absorbance change in the sample (ƛmax = 450 nm), which can then be measured using a plate reader (Infinite^®^ 200 Pro, Tecan, Männedorf, Switzerland) to determine the results for each sample. A D-lactate standard curve was established using a D-lactate standard prepared in serial dilution according to the manufacturer’s recommendations. Samples were assayed in duplicate, and the average result was compared to the standard curve. Samples outside the assay linearity range were diluted. Hemolytic serum samples were excluded from the analysis because hemolysis could cause interference in the D-lactate measurements when using a colorimetric assay kit.

### 2.3. Statistical Analysis

The descriptive analysis of the clinical and laboratory variables was reported using means, medians, and interval ranges for continuous variables and numbers and percentages for qualitative and semi-quantitative variables.

The results obtained from the serum D-lactate concentration assay were reported as means, medians, and interval ranges. Statistical significance was determined using an analysis of variance (ANOVA) with Tukey’s test as a post hoc test for multiple comparisons. This test was chosen after analysis of the entire data distribution using the Shapiro–Wilk normality test (*p* = 0.2744, V = 0.9577). A *p* value < 0.05 was considered statistically significant.

## 3. Results

### 3.1. Animals

A total of 18 dogs with IBD and 10 healthy control dogs were included in the study.

IBD Group: The most prevalent breeds in the IBD group were German Shepherds (9/18; 50%) and mongrels (3/18; 16.67%). Additionally, several other breeds were represented by few cases each: Golden Retriever (2/18; 11.11%), Pug (1/18; 5.56%), Cocker Spaniel (1/18; 5.56%), Pincher (1/18; 5.56%), and Weimaraner (1/18; 5.56%). Six animals (33.33%) were working dogs, whereas twelve dogs (66.67%) were companion animals. There were 10 females (55.56%) and 8 males (44.44%), with 4 neutered females and 1 castrated male. The mean age was 5.72 years (median, 5.5 years; range, 2–11 years), and the mean body weight was 21.12 kg (median, 23 kg; range, 4–27.4 kg). The mean serum concentration Alb was 2.49 g/dL (median, 2.5 g/dL; range, 1.05–3.61 g/dL), and the mean serum concentration of total protein (TP) was 5.7 g/dL (median, 6 g/dL; range, 2.9–7.3 g/dL). Table 2 summarizes the signalment, the albumin and total protein concentrations, the WSAVA histopathological grading for each section of the GI tract, and the CCECAI by IBD group. The mean score of the CCECAI was 5.83 (median, 5; range, 0–18). Five dogs (27.78%) were classified as having severe IBD, with a mean “Index for IBD activity” score for this group of 7.6 (median, 7; range, 7–9). Among these dogs, four (80%) with severe IBD died, while only one (20%) survived. Eight dogs (44.44%) were classified as having moderate IBD, with a mean “Index for IBD activity” score for this group of 4.37 (median, 4; range, 4–5). Five dogs (27.78%) were classified as having mild IBD, with a mean “Index for IBD activity” score for this group of 2.8 (median, 3; range, 2–3). The DI was available for only one dog, with a DI of 4.5 (normal DI < 0), indicating a reduction in *C. hiranonis* and an increase in *E. coli*.

Control Group: The most prevalent breeds in the control group were German Shepherds (6/10; 60%) and mongrels (2/10; 20%). Four animals (40%) were working dogs, whereas six (60%) dogs were companion animals. Various other breeds were also reported in this group, with a prevalence of 10% (1/10 dogs) each, including Weimaraner, Labrador Retriever, and Bernese Mountain dogs. The sex distribution within the groups was homogeneous, with 50% male and 50% female (5/10 each); there were no neutered animals. The mean age was 5.1 years (median, 4 years; range, 1–14 years), and the mean body weight was 30.16 kg (median, 29.5 kg; range, 22–35.6 kg). Table 3 summarizes the signalment for the control group.

### 3.2. D-Lactate Concentrations

The mean D-lactate concentration of the IBD group was 223.59 μM (median, 259.58 μM; range, 66.53–386.56 μM). The mean D-lactate concentration of the group with severe IBD was 196.76 μM (median, 174.89 μM; range, 88.20–374.64 μM). The mean D-lactate concentration of the group with moderate IBD was 245.58 μM (median, 268.35 μM; range, 99.79–386.56 μM). The mean D-lactate concentration of the group with mild IBD was 215.22 μM (median, 254.53 μM; range, 66.53–373.66 μM). Table 2 summarizes the concentrations of serum D-lactate for the IBD group. The mean D-lactate concentration of the control group was 209.77 μM (median, 208.17 μM; range, 139.77–300.50 μM). Table 3 summarizes the concentrations of serum D-lactate for the control group. No significant differences for the D-lactate concentrations were identified (*p* > 0.05 for each) between the control group and the IBD group nor within the IBD group related to severity of the disease (Figure 1).

## 4. Discussion

To our knowledge, this is the first study to investigate the serum D-lactate concentrations in dogs with IBD. Our results show no difference in the serum concentrations of D-lactate between dogs with various degrees of disease severity (severe, moderate, mild) and healthy dogs.

The pathogenesis of IBD in dogs is poorly understood, although evidence from human and animal models suggests the primary role of the intestinal microbiota in influencing aberrant immunological host responses [23]. The gut microbiota plays an essential role in maintaining host health, and there is growing evidence of the relationship between the metabolites produced by the intestinal microbiota and health status in both humans and animals [24,25]. Although D-lactate dehydrogenase can synthesize D-lactate from pyruvate and has recently been identified in mammalian mitochondria, its enzymatic activity does not seem to significantly impact the endogenous metabolism of D-lactate in these animals [26]. For these reasons, in mammalian serum, D-lactate primarily originates from exogenous sources, such as carbohydrates undergoing anaerobic fermentation by intestinal bacteria, and is subsequently absorbed in the intestine [2]. Bacterial species capable of producing D-lactate, which is subsequently absorbed by the intestinal mucosa, are commonly found in the colon. These bacteria require the enzyme D-lactate dehydrogenase for metabolite production [2,27]. The systemic absorption of D-lactate in dogs is mediated by the presence of MCT1 in the colon [3,10]. Recent studies have identified several altered metabolites in the feces of dogs with CEs, including D-lactate [10,17]. Given the recent importance attributed to intestinal dysbiosis as a trigger or consequence of canine IBD, along with the observed increase in fecal D-lactate concentrations in dogs with CEs, increased serum D-lactate concentrations could be associated with IBD [21,23,28]. Our results show highly variable D-lactate concentrations in both healthy and IBD groups. No significant differences were observed in the serum D-lactate concentrations between the two groups or among dogs with IBD of different severity levels. However, some dogs in the IBD group exhibited extremely low serum D-lactate concentrations, while others showed very high concentrations. Although the range of D-lactate concentrations was similar in the healthy dogs (139.77–300.50 µM) and the IBD dogs (66.53–386.56 µM), the healthy dogs never exhibited such extreme values. Previous studies on D-lactate as a single microbial metabolite have shown that its concentrations can increase in feces while remaining normal in the serum in human patients [27,29,30]. Similarly, a study in veterinary medicine that examined the fecal microbiota and serum metabolite profiles in dogs with IBD found few significantly altered metabolites in the serum despite alterations in several major bacterial groups [31]. This study, along with our results, suggests that some as-yet unidentified factors could influence the absorption of D-lactate and other metabolites in dogs with IBD. With the systemic absorption of D-lactate mediated by the presence of MCT1 in the colon, a decrease in MCT1 expression could influence D-lactate absorption and needs further evaluation.

Our study identified a significant difference in the D-lactate concentrations compared to previous studies. Nappert et al. (2002) reported higher concentrations for both dogs with parvoviral enteritis (CPV) (2350 ± 2760 μM) and healthy dogs (2690 ± 1830 μM) [11]. This difference can be explained by the different methods used to measure the serum D-lactate concentrations: our study utilized a colorimetric assay kit involving the enzymatic oxidation of D-lactate to determine the serum concentrations, whereas Nappert et al. (2002) [11] used high-performance liquid chromatography. The difference in the serum D-lactate concentrations can also be associated with the sample storage conditions. In our study, the serum samples were stored at −80 °C, whereas Nappert et al. (2002) [11] stored their serum samples at −20 °C. Conversely, a previously conducted study by Venn et al. (2020) [3] compared the serum D-lactate concentrations in healthy and CPV-infected dogs, showing D-lactate concentrations like those in our study. Venn et al. (2020) [3] reported concentrations for dogs with CPV of 469 ± 173 μM and for healthy dogs of 306 ± 45 μM [3]. This study utilized the same colorimetric methods used in our study. Other studies have identified significant differences in the serum lactate concentrations in dogs compared to our study, but these studies have typically measured the total lactate or L-lactate concentrations [32,33]. The median D-lactate concentrations obtained in our population of IBD and healthy dogs (223 μM; range, 66–386 μM; 209 μM; range, 139–300 μM) was different from the concentrations observed in cats with gastrointestinal disease and healthy cats (360 μM; range, 40–8330 μM; 220 μM; range, 40–870 μM) [9]. Whether this is a species-specific difference remains undetermined.

Lactate-producing bacteria in the gut are often considered useful, as they can reduce luminal pH and help counteract pathogens. However, conditions associated with maldigestion or excessive carbohydrate fermentation can create a favorable environment for the further proliferation of acid-resistant D-lactate-producing bacteria [3,10]. In human patients with short bowel syndrome and diarrheic calves with D-lactate acidosis, studies have detected an increase in lactate-producing bacteria, including *Lactobacillus*, *Streptococcus*, *Veillonella*, *Ligilactobacillus*, and *Olsenella*. Additionally, lower abundance and diversity have been observed in the gut microbiota during GI disease in both humans and animals [3,10,21]. In a study by Blake et al. (2016) [25], dogs with EPI were found to have a significantly lower abundance of *Fusobacterium* and *C. hiranonis* and significantly higher numbers of *E. coli*, *Lactobacillus*, and *Bifidobacterium* compared to healthy dogs. Additionally, both dogs with EPI and CEs have exhibited significantly increased fecal D-lactate concentrations [10]. Similarly, in dogs with CEs, significant differences were observed in their gut microbiota composition compared to that of healthy dogs. Specifically, the total bacteria, *Faecalibacterium*, *Turicibacter*, *Blautia*, *Fusobacterium*, and *C. hiranonis* levels were significantly lower in dogs with CEs, whereas the *E. coli* and *Streptococcus* levels were significantly higher. These microbiome changes can be easily evaluated using a dysbiosis index (DI) [19]. These studies suggest that the composition of the gut microbiota significantly influences the concentration and production of D-lactate, and vice versa [2,10,19,27]. In our study, the DI was available for only one dog with IBD, revealing severe dysbiosis with a DI of 4.5. This dog exhibited a reduction in the concentration of *C. hiranonis* and an increase in the percentage of *E. coli*. Notably, this dog also had moderately high serum D-lactate concentrations (272.07 µM), suggesting a potential association with increased GI D-lactate production and absorption through the epithelial barrier. Further studies are needed to assess the correlation between D-lactate concentrations and the DI.

The prevalent breeds represented in both the IBD and control groups were German Shepherds (50% and 60%, respectively) and mongrels (16.65% and 20%, respectively). The current literature suggests an increased susceptibility to IBD in German Shepherds, which could be related to genetic factors. Specifically, German Shepherds have been reported to exhibit a low expression of Toll-like receptor 4 (TLR4) and Toll-like receptor 5 (TLR5), which are responsible for activating the innate immune system [34,35,36,37]. In our study, all the dogs with IBD exhibited chronic GI signs, with an average CCECAI score of 5.83. In contrast to previous reports involving cats and humans with GI signs and increased concentrations of D-lactate, our study did not observe neurological symptomatology in dogs with IBD [3,6].

A multitude of clinical, laboratory, and histopathologic variables have been proposed as useful indices to characterize disease activity in dogs affected by IBD [12,22,25,26,27,28]. Several laboratory parameters such as cobalamin, folate, C-reactive protein, serum Alb, canine pancreatic lipase immunoreactivity (cLPI), and fecal calprotectin have been evaluated as predictive and prognostic markers in canine IBD [8,17,38,39,40,41]. In particular, serum Alb and TP concentrations, along with histopathology grading, were evaluated for their potential usefulness in predicting the outcome of IBD in dogs [22]. Hypoalbuminemia has previously been reported to be associated with treatment refractoriness in IBD. Additionally, the CCECAI was shown to predict the severity and negative outcomes of the disease in a large prospective study [34,41]. Notably, the CCECAI is composed of numerous clinicopathologic variables, including serum Alb concentrations. However, it does not incorporate the degree of mucosal inflammation detected at the histological level. Additionally, in our experience, some dogs with IBD do not show typical GI signs but rather ascites and/or peripheral edema as a consequence of protein loss in the GI tract and hypoalbuminemia. Consequently, we suggest that the CCECAI score cannot accurately reflect the severity of the disease in these cases. To overcome this limitation, we introduced an “Index for IBD activity” in this study to comprehensively assess the clinical activity of the disease. Although albumin concentrations are already considered in the CCECAI scoring system, we have categorized this separately due to the prognostic value of this parameter in dogs affected by IBD. In our opinion, this could avoid underestimating the severity of IBD in dogs with hypoalbuminemia but with a normal or mild CCECAI score. Our proposed index also considers the severity of histological changes and could be useful as a prognostic index, considering that all the deceased dogs belonged to the severe IBD group. However, further validation studies with a large cohort of dogs are needed to assess the prognostic utility of the “Index for IBD activity”.

Our study has several limitations. The main limitations of the study are related to its retrospective nature and the small number of animals included. Another limitation of the present study is related to the method used to measure the serum D-lactate concentrations. Our study utilized a colorimetric assay kit, whereas most previous studies have used high-performance liquid chromatography. Moreover, the colorimetric assay kit has not been validated for dogs. However, this method was already used in a previous study in dogs affected by parvoviral enteritis, with similar results [3]. The lack of homogeneity of the study population is another limitation. Specifically, the dogs affected by IBD showed a lower body weight compared to the control group. However, it is necessary to note that healthy dogs, being part of our blood donor program, have weight requirements for donation. Moreover, although the control group showed no clinical signs of GI disease and had no hematological changes at the screening exams, they were not tested for gastroenteric diseases by endoscopy. Finally, no dog except one had data on its gut microbiome. Further studies should assess the serum D-lactate concentrations in healthy dogs with a normal DI and compare them with dogs affected by IBD and/or other CEs such as FRE, ARE, IRE, or NRE.

## 5. Conclusions

The aim of the present study was to measure the serum D-lactate concentrations in dogs with IBD. No statistical difference was observed in the serum concentrations of D-lactate between dogs with various degrees of IBD and healthy dogs, although a wide spectrum of D-lactate concentrations was observed in the IBD dogs. This aspect suggests further evaluations on this topic to understand potential factors able to influence the serum D-lactate concentrations in dogs affected by IBD.

## Figures and Tables

**Figure 1 animals-14-01704-f001:**
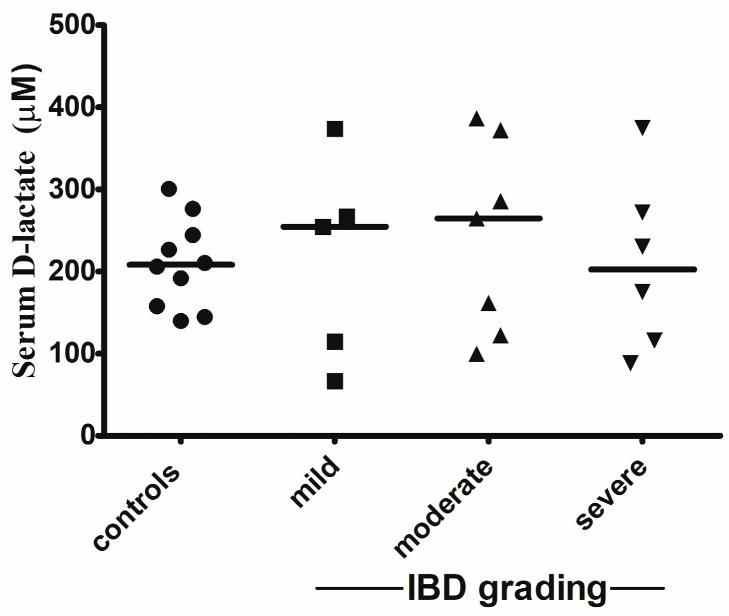
This figure shows individual concentrations and group medians (line) for the D-lactate concentrations in the IBD (severe, moderate, and mild) and control groups.

**Table 1 animals-14-01704-t001:** Scoring “Index for IBD activity” based on CCECAI, WSAVA histopathological grading, and concentrations of serum Alb.

Index for IBD Activity	
CCECAI	0 = Normal CCECAI (0–3)
	1 = Mild CCECAI (4–5)
	2 = Moderate CCECAI (6–7)
	3 = Severe CCECAI (9–11)
	4 = Very Severe CCECAI (>12)
WSAVA histopathological grading	0 = Normal
	1 = Mild
	2 = Moderate
	3 = Marked
Alb concentrations	0 = > 2 mg/dL
	1 = 1.5–1.9 mg/dL
	2 = 1.2–1.49 mg/dL
	3 = < 1.2 mg/dL
Total Score (Mild IBD ≤ 3); Moderate IBD (4–6); Severe IBD (≥7)	

Albumin (Alb); Canine Chronic Enteropathy Activity Index (CCECAI); inflammatory bowel disease (IBD); World Small Animal Veterinary Association (WSAVA).

**Table 2 animals-14-01704-t002:** Signalment, albumin (Alb) and total protein (TP) concentrations, WSAVA histopathological grading for each section of GI tract, CCECAI, outcome, Index for IBD activity, and concentrations of D-lactate for each dog in the IBD group (severe, moderate, and mild).

Signalment	Alb g/dL	TP g/dL	WSAVA Histological Grading, Stomach/Duodenum/Ileum/Colon	CCECAI	Outcome	Index for IBD Activity	D-Lactate (μM)
Pug, NF, 8 yrs, 6 Kg	1.05	2.9	Normal/Marked/Marked/Mild	18	Dead	9	374.65
GS, NF, 9 yrs, 18 Kg	1.18	3.6	Moderate/Moderate/Moderate/Moderate	13	Dead	8	115.94
GS, M, 5 yrs, 26 Kg	1.9	5.7	Mild/Marked/Marked/Moderate	6		7	174.90
Cocker Spaniel, M, 10 yrs, 7 Kg	1.7	4.8	Mild/Marked/Marked/Mild	14	Dead	7	230.16
Mongrel, F, 3 yrs, 18.7 Kg	1.60	3.1	Mild/Marked/Moderate/Mild	10	Dead	7	88.20
GS, F, 6 yrs, 27.4 Kg	2.57	5.62	Mild/Marked/Moderate/Mild	4		5	272.07
GS, M, 2 yrs, 26.9 Kg	2.4	5.5	Normal/Moderate/Moderate/Mild	6		5	285.60
GS, M, 2 yrs, 18.5 Kg	3.10	6.20	Mild/Moderate/Moderate/Mild	6		5	122.41
Golden Retriever, NF, 9 yrs, 24 Kg	3.2	7.1	Marked/Marked/Moderate/Mild	0		4	264.64
GS, F, 3 yrs, 15.3 Kg	2.99	5.8	Mild/Moderate/Mild/Mild	4		4	372.03
Mongrel, M, 3 yrs, 22 Kg	3.61	6.9	Normal/Moderate/Mild/Mild	4		4	161.58
Golden Retriever, F, 6 yrs, 35 Kg	3.49	6.6	Normal/Moderate/Mild/Mild	4		4	99.80
Mongrel, CM, 5 yrs, 4 Kg	2.5	6.4	Mild/Moderate/Mild/Mild	7		4	386.56
GS, F, 5 yrs, 26 Kg	2.68	6.5	Moderate/Moderate/Mild/Mild	3		3	373.67
Weimaraner, NF, 6 yrs, 31 Kg	3.0	7.3	Mild/Moderate/Mild/Mild	2		3	266.83
GS, M, 7 yrs, 38 Kg	2.4	5.5	Mild/Moderate/Mild/Mild	1		3	114.53
Pinscher, F, 11 yrs, 6.4 Kg	2.51	6.5	Normal/Moderate/Mild/Mild	0		3	254.53
GS, M, 3 yrs, 30 Kg	3.04	6.71	Normal/Mild/Mild/Mild	3		2	66.54

Albumin (Alb), Canine Chronic Enteropathy Activity Index (CCECAI), castrated male (CM), female (F), German Shepherd (GS), inflammatory bowel disease (IBD), years (yrs), male (M), neutered female (NF), total protein (TP), World Small Animal Veterinary Association (WSAVA).

**Table 3 animals-14-01704-t003:** Signalment and concentrations of D-lactate for control group.

Signalment	D-Lactate (μM)
Weimaraner, M, 8 yrs, 38 Kg	191.75
Labrador Retriever, F, 6 yrs, 24 Kg	276.20
GS, F, 14 yrs, 35.6 Kg	210.36
Mongrel, M, 7 yrs, 25 Kg	139.78
Mongrel, M, 1 yrs, 22 Kg	144.62
Bernese Mountain dogs, M, 3 yrs, 50 Kg	244.33
GS, F, 3 yrs, 28.5 Kg	205.99
GS, F, 2 yrs, 22 Kg	226.45
GS, F, 5 yrs, 31.5 Kg	300.50
GS, M, 2 yrs, 30.5 Kg	157.81

Female (F), German Shepherd (GS), years (yrs), male (M).

## Data Availability

The original contributions presented in the study are included in the article; further inquiries can be directed to the corresponding authors.

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
