# Peer review of "Serum D-Lactate Concentrations in Dogs with Inflammatory Bowel Disease"

_animals, 2024, doi:10.3390/ani14111704_

Round 1
Reviewer 1 Report
Comments and Suggestions for Authors
Please see attached document for comments and suggestions.

Comments on the Quality of English LanguagePlease see attached document for comments and suggestions.
Author Response
We thank the Reviewer for her/his insightful comments and suggestions. We have addressed all the comments raised and we hope that our changes and responses below will satisfy the Reviewer.
Reviewer: The manuscript entitled “Serum D-lactate concentrations in dogs with inflammatory bowel disease” provides new information to the veterinary research community about D-lactate in dogs with IBD. Although the D-lactate concentrations were not different between groups it is interesting and relevant information that the range of concentrations was highly variable. This can help guide future research in this topic, by allowing for more accurate and useful power calculations for future study sample sizes. The manuscript has numerous grammatical and spelling errors as well as inconsistencies in the results and methods sections. It is unclear what the units of concentration are for the D-lactate, which is a crucial piece of information. Regardless, the concentrations found are either much higher or lower than results from similar studies. The literature review in introduction and discussion sections could be much improved, by discussion of concentrations of D-lactate in other studies and how they relate to this study. The assay kit used is not specific for canine serum testing and therefore should include some amount of validation testing to ensure fit-for-purpose. Additionally, the new scoring system the authors implemented for scoring IBD in dogs is unclear in how it was calculated and does not match the results. With heavy editing of the manuscript, and improved clarity of key aspects, it may be a useful addition to the veterinary research community.
Authors: We appreciate the Reviewer for her/his excellent suggestions. The unit of measurement used for D-lactate is μM, as reported in the Abstract and Materials and Methods. Therefore, we have amended the manuscript if typos were present. Following the Reviewer’s suggestion, we have improved the literature review in the Introduction and Discussion with other studies that investigated D-lactate concentrations in serum/feces and microbiome changes in humans and animals. Our study utilized a colorimetric assay kit to determine serum D-lactate concentrations as reported in a previous study in dogs (Veen et al.2020); however, as Reviewer suggests, the assay kit used is not specific for canine serum and this can be a limitation. We have added this in the discussion.
R: Please see the following points for specific comments:
- Scoring Index for IBD:
- The CCECAI scoring system takes into account albumin levels. Therefore, the inclusion of albumin level as another separate category is confusing because it seems more weight is being placed on albumin level.
- The authors do not define what constitutes as normal/mild/moderate/severe CCECAI or WSAVA scoring. For CCECAI (according to Allenspach citation), normal score is 0-3, mild is 4-5, moderate is 6-8, severe is 9-11, and very severe is 12 or higher. For WSAVA histopathological scoring, were multiple tissues biopsied? Normally, a score is given to each section, like antrum, duodenum, colon, etc. Was this an average of all histo scores obtained, or the maximum score? Please define what initial CCECAI and WSAVA scores translated to in the IBD scoring system.
- When discussing results of the IBD scoring index, the authors use the term “clinical score”. This is confusing because it is not referred to as “clinical score” prior to the results. A consistent terminology for the new scoring system should be used throughout.
- In table 1, a score of 3 is defined as moderate IBD. However, in the results, the scores of 3 were placed in the mild group. Please make corrections to be consistent.
A: Scoring Index for IBD
- Although albumin level is already considered in the CCECAI scoring system, we have categorized this separately due to the prognostic value of this parameter in dogs affected by IBD. Based on our experience, some dogs with IBD don’t exhibit gastrointestinal signs and owners reported abdominal distention o peripheral edema due to hypoalbuminemia as primary clinical signs. Therefore, we have tried to overcome this limitation in our proposed “Index for IBD activity”. This is the reason for using of the albumin level as a separate variable in our proposed index. However, if the Reviewer does not find our response to her/his concerns satisfactory, we can try to change our “Index for IBD activity” by removing albumin levels as a separate index.
- We have defined what constitutes normal, mild, moderate, and severe CCECAI according to the criteria established by Allenspach et al. and we have amended the Materials and Methods. Concerning WSAVA histopathological grading, we considered the maximum inflammatory score observed in various sections (stomach, duodenum, colon, and ileum) of the gastrointestinal tract and we have amended the Materials and Methods.
- We have changed the term “clinical score” with “Index for IBD activity” in the Results.
- Materials and Methods and tables have been modified to be consistent with the Results.
R: D-lactate kit and results:
- What was the concentration range of the standard curve (0-200nmol/mlinprotocol from manufacturer)?
- What was the LLOQ and ULOQ? Precision and accuracy of standards? Repeatability of samples? Was any amount of validation performed with this assay assuming it is the first time being used in canine serum samples and has not been previously validated?
- Were any samples hemolytic or lipemic, and if so, was that type of sample validated for use with this kit?
- Did samples need to be deproteinized or have any pretreatment or dilution to them? If so, please briefly describe the procedure.
- D-lactate concentrations are reported in several different ways. In the results section, they are listed as μM/ml. On the figure 1 y-axis, it is listed as μ It should be “μmol/mL” or “μM” in all instances depending on the assay results. The two do not mean the same thing.
- Concentrations in table 2 are listed to 8 decimal places. Typically, concentrations should only include as many decimal places as are validated. For example, if the assay cannot differentiate picomole levels of D-lactate, but can differentiate single digit nanomole levels, then only 3 decimal places should be included.
- Figure 1 would be more helpful if showing the individual sample points as well as medians (because of non-normal distribution).
- Concentrations found in this study need to be re-evaluated as they are much higher/lower than any listed in the literature, unless the units of measurement were reported incorrectly. Caines et al., Can J Vet Res, 2013. listed upper limit of plasma lactate as 2.5 mM. Blake et al., PloS One, 2019 listed fecal total lactate between 0 and 75 mM. de Papp et al., J Am Vet Med Assoc, 1999, plasma lactate median in dogs with GDV was 6.6 mM. Nappert et al., Canadian Journal of Vet Res, 2002, listed mean D- lactate in control dogs as 2.69 mM and mean D-lactate in parvo puppies as 2.35 mM.
A: D-lactate kit and results
- We have used ranges reported in the kit (0-200 nmol/ml as protocol from manufacturer).
- Samples have been diluted to fit within the linearity range. All out-of-range samples were further tested after dilution. Some samples were diluted to allow it to fall within the linearity range. The LLOQ was 0 while the ULOQ was 200. The linearity range between all tested concentrations is between 0 - 200 in fact between these concentrations the R2982. We have used a colorimetric assay kit previously used by Venn et al. (2020). We recognize that this is a limitation of our study, so we have added this in the discussion and included this as a limitation.
- Hemolysis can cause interference in D-lactate measurement when using a colorimetric assay for this reason hemolytic serum samples were not included in the study. We have added this information in the Materials and Methods. There were not lipemic samples.
- No samples needed pretreatment or deproteinization. Some samples were diluted to allow it to fall within the linearity range.
- The unit of measurement used for D-lactate is μM. We have amended the manuscript.
- Concentrations in Table was modified using 2 decimal places.
- The graphic has been changed as required.
- Concentrations found in our study are similar with concentrations of D-lactate found in a previous paper (Venn, E.C.; Barnes, A.J.; Hansen, R.J.; Boscan, P.L.; Twedt, D.C.; Sullivan, L.A. Serum D-lactate concentrations in dogs with parvoviral enteritis. Vet. Intern. Med. 2020; 34(2), 691-699) on dogs with parvoviral enteritis. In this previous study has been used the same colorimetric kit for determination of level of D-lactate. We have discussed this in Discussion.
R: Statistical analysis:
For comparison of three or more groups, ANOVA assumes normal distribution. Did the data have normal distribution (what method to test distribution of data)? Assuming it is non-normally distributed based simply on number of samples in each group, comparison of three or more groups should use Kruskal Wallis testing. For comparison of two groups (if comparing all IBD v HC), with non-normal distribution, Wilcoxon test should be used.
A: The distribution of the data was evaluated by Shapiro Wilk Normality test. This test evidenced the normal distribution of all data (p=02744, V=0,9577).
R: Discussion:
- D-lactate has been measured in feces of dogs with chronic enteropathy and semi-correlated with lactate producing bacteria abundances (Blake et al., PLoS One 2019). Results should be discussed in relation to the findings in the current literature, including the findings of D-lactate in cats with GI disease. How did the concentrations compare between dogs and cats?
- If concentrations found in this study are actually 66-386 μM, then the authors need to explain potential reasons for the much lower concentrations found in relation to previous studies. Possibly storage of samples, sampling technique, etc., with references.
- Line 235-237: Not a complete sentence. But also, confusing why the authors would say that albumin reduction is not necessarily indicative of protein loss, but then include it twice in the IBD scoring system.
- Line239: “proven useful as a prognostic index” is extremely strong conclusions based on the little data available. It could be a useful prognostic index, but validation testing with many more dogs and training and testing cohorts with sophisticated statistics would need to be done.
A: Discussion
- The discussion has been improved considering previous published literature, including the correlations with lactate-producing bacterial abundances or dysbiosis, and the concentration of D-lactate in cats with GI diseases and healthy individuals.
- The different concentration reported in our study respect to previous studies has been discussed in the Discussion.
- Line 235-237 & 239: We modified the paragraph with the hope that it is clearer.
- We have amended the sentence following the Reviewer’s suggestion.
R: Limitation
That healthy controls did not have GI testing.
A: We have added that healthy controls did not undergo GI testing as a limitation.
R: Line 43: “For this these reasons...”
Line 192: “For this reasons...” is used multiple times and is grammatically incorrect. “reasons” is plural and therefore needs “these”
A: Line 43 and in all Text: We have amended with “For these reasons..”
R: Line 55-56: “...marked by persistent or recurrent GI signs lasting, and from a clinical perspective...”
A: Line 55-56: We have amended as suggested by Reviewer.
R: Line 67: “index dysbiosis” should be “dysbiosis index”
A: Line 67: Done
R: Line 68: “the available” should be “the availability”
A: Line 68: Done
R: Line 85: Several words are misspelled. Should be complete blood count and serum (or plasma) biochemical profile.
A: Line 85: Done
R: Line 87: “dietetic” is not commonly used. More often “diet trial” or “dietary modification trial” & Line 87: “for at least of 4 weeks”
A: Line 87: We have changed the sentence with “diet trial” and “for at least of 4 weeks”
R: Line 88: What was the dose or range of doses for the probiotic?
A: Line 88: The dose for the probiotic is one packet (4.4 g) q24h. We have added this information in the manuscript.
R: Line 94: Grammatically incorrect. “Dogs with IBD were divided into three subcategories...”
A: Line 94: We have amended as suggested by Reviewer.
R: Line 98: Reword this sentence to make more clear.
A: Line 98: Sentence was reworded.
R: Line 137: “... was considered...”
A: Line 137: We have amended as suggested by Reviewer.
R: Line 155: “...score for this groups of...”
A: Line 155: We have amended as suggested by Reviewer.
R: Line 193: Reword this sentence to make it grammatically correct. Not sure what is meant the way it reads now.
A: Line 193: Sentence was reworded.
Reviewer 2 Report
Comments and Suggestions for Authors
The article is original and could be very relevant for the field. The aim of the study was to measure serum D-lactate concentrations in dogs with inflammatory bowel disease (IBD), to establish the relevance for diagnosis.
The results showed no significant difference (p > 0.05) in the serum levels of D-lactate between dogs with various degrees of IBD and healthy dogs.
The methology of the study should be improved with some imagistic or histopathologic aspects to justify how the IBD group was divided into three subcategories based on the severity of the disease.
The references are appropriate, including some relevant authors experience in the field.
I recommend some supplementary corrections.
Lines 140-166 data should be included in a Table or Fig, to be easier readable
Author Response
We thank the Reviewer for her/his insightful comments and suggestions. We have addressed all the comments raised and we hope that our changes and responses below will satisfy the Reviewer.
Reviewer: The article is original and could be very relevant for the field. The aim of the study was to measure serum D-lactate concentrations in dogs with inflammatory bowel disease (IBD), to establish the relevance for diagnosis. The results showed no significant difference (p > 0.05) in the serum levels of D-lactate between dogs with various degrees of IBD and healthy dogs. The methodology of the study should be improved with some imagistic or histopathologic aspects to justify how the IBD group was divided into three subcategories based on the severity of the disease. The references are appropriate, including some relevant authors experience in the field. I recommend some supplementary corrections.
Lines 140-166 data should be included in a Table or Fig, to be easier readable
Authors: We appreciate the Reviewer for her/his excellent suggestions. The Scoring Index for IBD uses the WSAVA histopathologic grade as the maximum inflammatory score observed in various sections (stomach, duodenum, colon, and ileum) of the gastrointestinal tract to establish three subcategories for the severity of IBD. Data one signalment, Alb e TP level, WSAVA Histopathological grading for each portion of GI tract, CCECAI, outcome, Index for IBD activity, and level of D-lactate for each dog with IBD was included in a Table 2. Data one signalment and level of D-lactate for each healthy dog was included in a Table 3.
Round 2
Reviewer 2 Report
Comments and Suggestions for Authors
The authors have made substantial improvements of the article. I recommend the publication in actual revised version
Author Response
We thank the Reviewer for her/his help during the review process.